# Bulk Photovoltaic Current Mechanisms in All-Inorganic Perovskite Multiferroic Materials

**DOI:** 10.3390/nano13030429

**Published:** 2023-01-20

**Authors:** Jiazheng Chen, Guobin Ma, Boxiang Gong, Chaoyong Deng, Min Zhang, Kaixin Guo, Ruirui Cui, Yunkai Wu, Menglan Lv, Xu Wang

**Affiliations:** 1Key Laboratory of Functional Composite Materials of Guizhou Province, College of Big Data and Information Engineering, Guizhou University, Guizhou 550025, China; 2School of Electronics and Information Engineering, Guiyang University, Guiyang 550005, China; 3Guiyang Makers Center, Guizhou 550025, China; 4School of Chemistry and Chemical Engineering, Guizhou University, Guizhou 550025, China

**Keywords:** all-inorganic perovskite, bulk photovoltaic, shift current, multiferroic

## Abstract

After the discovery of bulk photovoltaic effect more than half a century ago, ferro-electrical and magneto-optical experiments have provided insights into various related topics, revealing above bandgap open voltages and non-central symmetrical current mechanisms. However, the nature of the photon-generated carriers responses and their microscopic mechanisms remain unclear. Here, all-inorganic perovskite Bi${}_{0.85}$Gd${}_{0.15}$Fe${}_{1-x}$Mn${}_{x}$O${}_{3}$ thin films were prepared by a sol-gel process and the effects of Gd and Mn co-doped bismuth ferrites on their microtopography, grain boundries, multiferroic, and optical properties were studied. We discovered a simple “proof of principle” type new method that by one-step measuring the leakage current, one can demonstrate the value of photo generated current being the sum of ballistic current and shift current, which are combined to form the so-called bulk photovoltaic current, and can be related to the prototype intrinsic properties such as magneto-optical coupling and ferroelectric polarization. This result has significant potential influence on design principles for engineering multiferroic optoelectronic devices and future photovoltaic industry development.

## 1. Introduction

Since the 1970s, the related research on ferroelectric photovoltaic has been carried out along with the research on the optoelectronic properties of ferroelectric materials. Ferroelectric is characterized by spontaneous polarization due to its central symmetry breaking [1]. Compared with ordinary dipole polarization, ferroelectric materials are always in a polarized state. When the applied electric field is greater than its coercive field, the polarization state can be switched, so that it can be regulated by one field between two or more states [2]. As early as half a century ago, Chynoweth found that barium titanate could produce stable photocurrent after absorbing light, which was later called “bulk photovoltaic effect” (BPE) [3]. It is generally believed that BPE originates from the spontaneous polarization of ferroelectrics generated by non-centrosymmetric structure. The carriers excited in real space are accompanied by “virtual” displacement. The imbalance of potential barrier will cause the carriers in one direction to be scattered, and the carriers in the other direction will move to the barrier of lower energy level. Such net current is called “shift current” $j_{sh}$[4]. Another type of current in BPE is called “ballistic current” $j_{b}$, which is caused by the asymmetry of photo-excited carrier momentum from the Fermi-Dirac distribution, and can be measured by the Hall effect [5]. In traditional semiconductor p-n junction solar cells, the position of quasi Fermi level limits that the energy difference of electron hole pair in thermal equilibrium state that cannot exceed its bandgap (the Shockley and Queisser limit) (SQL) [6]. From this point of view, although the specific microscopic mechanism of BPE is still controversial [7], it must be different from the traditional p-n junction space depletion layer, because both theoretical calculation and experimental results show that [8,9], the open circuit voltage (Voc) of ferroelectric materials (or localized polar materials) [10] can exceed the limit of SQL, reaching two to four orders of magnitude higher than the bandgap [11], which enables ferroelectric photovoltaic devices with significant potential to be applied in multifunctional optical–electrical systems [12].

Perovskite solar cells (PSCs) are generally defined by a typical perovskite composition ABX${}_{3}$, so A and B are cations and X are anions with different charges and sizes [13]. Methylammonium lead halides perovskite is typical of this, where the CH${}_{3}$NH${}^{3+}$ cation is encircled by an octahedron of PbX${}_{6}$. The X ion (X = I, Br, Cl or O) is movable and can roam through the entire crystal. PSCs belong to the third generation of solar cells, and the earliest research on PSCs originated from dye-sensitized solar cells [14]. However, the device stability was rather poor due to the rapid dissolving of the perovskite materials in organic solvents. Subsequent explorations by many scientists have led to numerous improvements in aspects such as perovskite materials and device structures, thus pushing the research and enthusiasm related to PSCs to a higher level [15,16,17]. Among them, one of the most promising route is go all-inorganic, as extensive researches have shown that all-inorganic halide perovskites express superior intrinsic structural stability and much lower sensitivity against humidity environment [18]. The calculated power conversion efficiency (PCE) of single junction PSCs based on SQL is 33% and for instance, the mixed halide all-inorganic CsPbIBr${}_{2}$ perovskite PSCs have reached a PCE just exceeded 12% [19]. Researchers are taking various methods trying to close the PCE gap, such as reducing defect amount and improving carrier management [20]. The traditional ferroelectric perovskites may be able to play an alternative approach here, as they can empower a large Voc owing to the BPE, and they are also naturally all-inorganic.

Although the open circuit voltage can be as high as $10^{3}$–$10^{5}$ V/cm, ferroelectrics are typical dielectric materials with high dielectric constant and wide bandgap. The work function of interface between electrodes also affects their short circuit current [21]. Therefore, ferroelectric materials have not been able to climb on the top of the crown in the photovoltaic community. In recent years though, ferroelectric materials have been mentioned more as efforts done by researchers to continuously increase their photo induced current [22,23]. Moreover, they can be self-powered photodetectors and small-scale electricity generators simultaneously [24,25,26]. Thus to improve photocurrent while utilizing the above bandgap voltage, self-regulating ability and excellent stability of BPE in traditional oxide ferroelectric materials become an interesting topic with significant importance. However, it is still a grand challenge to separately measure shift current and ballistic current. BiFeO${}_{3}$ (BFO) is a typical single-phase multiferroic perovskite material with high Curie (T${}_{C}$ = 1103 K) and Néel (T${}_{N}$ = 643 K) transition temperatures. Local spin ordering and non-central symmetrical structural distortion protected magneto-electric coupling in BFO [27], made it an ideal candidate to study both ferroelectric photovoltaic effect and its connections with ferromagnetic in strongly correlated electron systems. This work successfully synthesized a serious of BFO dopant films with different concentrations of Gd and Mn, then the multiferroic and photovoltaic performances were analyzed in detail, exhibiting a brand new simplified method to classify the shift and ballistic current, and providing a feasible route of understanding the photo-induced current mechanism and thus realizing stable multiferroic all-inorganic perovskite solar cell devices.

## 2. Materials and Methods

Instead of using pure BFO to study the mechanisms. It is rather convinced that an effective improvement on the ferroelectric and piezoelectric performance can be made by ion substitution of lanthanide rare earth elements owing to modulating the phase structure [28]. Also, decreased leakage current density is found at high electric field for further promoting the ferroelectric performance in Mn-doped BFO films [29]. Besides, doping elements at B-site can modify the magnetic properties [30]. Therefore, BFO, Bi${}_{0.85}$Gd${}_{0.15}$FeO${}_{3}$ (BGFO), Bi${}_{0.85}$Gd${}_{0.15}$Fe${}_{1-x}$Mn${}_{x}$O${}_{3}$ (x = 0.04, 0.08, 0.12) (BGFMO-4, BGFMO-8, BGFMO-12, respectively) thin films were fabricated on Pt/Ti/SiO${}_{2}$/Si and ITO (Indium tin oxide)/glass substrates using a sol-gel chemical solution deposition method. Fe(NO${}_{3}$)${}_{3}$·9H${}_{2}$O, 99%, Bi(NO${}_{3}$)${}_{3}$·5H${}_{2}$O, 99%, Gd(NO${}_{3}$)${}_{3}$·6H${}_{2}$O, 99%, MnC${}_{4}$H${}_{6}$O${}_{4}$·4H${}_{2}$O are prepared as raw materials, with ethylene glycol methyl (C${}_{3}$H${}_{8}$O${}_{2}$) ether as the solvent, and citric acid (C${}_{6}$H${}_{8}$O${}_{7}$) as the chelating agent. Bi(NO${}_{3}$)${}_{3}$·5H${}_{2}$O, Fe(NO${}_{3}$)${}_{3}$·9H${}_{2}$O, GdN${}_{3}$O${}_{9}$·6H${}_{2}$O, and MnC${}_{4}$H${}_{6}$O${}_{4}$·4H${}_{2}$O are mixed in a certain amount of C${}_{3}$H${}_{8}$O${}_{2}$ at a stoichiometric ratio, heated and stirred (Bi excess 10%) till completely dissolved. Then, citric acid was added drop by drop while stirring in a water bath, and ethylene glycol methyl ether was added to the final volume, producing a rufous precursor 4 h later. The prepared precursor solution was filtered and aged for 48 h to obtain a sol. The aged sol was spin-coated on different substrates. Generally, a slow rotation of 10 s (500 rpm/min) followed by a rapid rotation of 30 s (4000 rpm/min) are used to make the sol coated uniformly on the surface of the substrates. After 200 ℃ baking and a rapid thermal processing at 400 ℃ for 300 s, the films were finally crystallized at 550 ℃ for 300 s. An array of Pt microelectrodes (Φ is of 0.5 mm) was prepared by sputtering and treated at 500 ℃ for 300s before testing.

The crystal structure was characterized by an X-ray Diffractometer (XRD, Rigaku SmartLab XG, Cu Ka radiation, voltage 40 kV, current 100 mA, and scanning speed 2${}^{\circ }$/min). The morphology and piezoelectricity were characterized respectively by a Scanning Electron Microscope (SEM , Hitachi SU8010) and an Atomic Force Microscope (AFM , Bruker MultiMode 8). The ferroelectric polarization as well as the leakage current were measured using a ferroelectric test system (Radiant Technologies Multiferroic 200 V) and the absorption spectrum was characterized by employing an integrating sphere (Hitachi U-4100). The magnetic hysteresis loop was measured by a physical property measurement system (PPMS Quantum Design). All tests were performed in room temperature.

## 3. Results

### 3.1. Structural Origin of Photocurrent

The as grown XRD patterns are shown in Figure 1a which fit in good agreement with the standard card (JCPDs No. 20-0169). Figure 1a confirms that pure BFO exhibits distorted perovskite structure with R3c space group [31]. While the samples incorporated with Gd and Mn ions experience peaks shifting towards larger angles as shown in the inset figure, where diffractions at (012) and (110) are combined with separation and merging upon Gd and Mn co-doping, indicating the existence of structural transformation. This transformation comes alongside with significant grain size decreasing from more than 150 nm to around 20 nm as exhibited in the following SEM images Figure 1b–f. Thin films shown in Figure 1 generally exhibit well-distinguished grains with clear gaps in between, whereas the surface morphology of Gd and Mn doped BFO turns to be dense and uniform with an average grain size of BGFO, BGFMO-4, BGFMO-8 and BGFMO-12 from 36.83 nm, 33.16 nm to 31.72 nm and 29.18 nm, respectively. The systematically uniformity and decrease of the grain size may be from two aspects: (i) The difference in ionic radius leads to lattices distortion that reduces the particle size and suppression of oxygen vacancy concentration. As a consequence, the grain growth rate becomes slower due to less oxygen ion carriers [32]. (ii) Doping of Gd${}^{3+}$ and Mn${}^{3+}$ can increase the number of nucleating centers assisting the nucleation rate and also the annealing rate as fast sequential-layer annealing can be activated [33]. Furthermore, the uniform fine grains and the less oxygen vacancies will contribute to the decrease in leakage current and enhancement in shift current in non-centrosymmetric materials, as discussed next.

The micro structure greatly affects the carrier distribution. The local PL lifetimes are studied to be shorter at grain boundaries probably caused by the presence of defect states or shallow trapping levels serving as nonradiative recombination centers [34]. In perovskites though studies have suggested that grain boundaries are less detrimental than in other semiconductors, or even beneficial [35]. Especially in ferroelectric materials, whose origin lies in the non-central symmetric brought domain walls, the ordering of defects at domain walls distributed along grain boundaries intriguingly affects their conductivity [36]. But how much degree of the structural asymmetry does affect the BPE photocurrent? What is more, the mechanisms of the BPE photocurrent are not trivial and have not been elucidated [37]. Therefore, it is rather practical to reconsider this problem from the theory of BPE, aiming at finding a simple method to display the components of photocurrent in these asymmetrical perovskite ferroelectrics.

### 3.2. Bulk Photovoltaic Current

In non-central symmetrical system, light absorption brings charge separation and results in the bulk photovoltaic current (BPC) with a form of:(1)$$\;j_{i}=j_{L}+j_{C}=G_{ijl}^{L}e_{j}e_{l}^{*}I+iG_{il}^{C}{\left[ee^{*}\right]}_{l}I$$

This phenomenon is called the BPE and it is sensitized to the polarization of the incident light [38]. Therefore the first part of Equation (Equation 1) is the linear BPC, related to linearly polarized light, and the second part is the circular BPC, corresponding to circularly polarized light. *I* is the light intensity, $e_{j}$ and $e_{l}^{*}$ are the components of the incident light polarization vector, $i[ee^{*}]$=σ$\vec{\kappa }/\kappa$ determines the degree of circular polarization σ, where κ is the photon wave vector, $G_{ijl}^{L}$ and $G_{il}^{C}$ are respectively third and second rank of photovoltaic tensors [4]. It has been presumed that circularly polarized light can only contribute to ballistic current [39]. As a result the ballistic current, under a electron-phonon interaction impurity band transition, can be simply expressed as:(2)$$\;j_{b}\approx j_{C}+j_{lb}$$
where $j_{lb}$ is the ballistic current under linearly polarized light. The “≈” sign here indicates there might be other current components for instance a $j_{H}$ under a built-in magnetic field. However, the linearly polarized light excited BPC is the sum of ballistic and shift components [5] so:(3)$$\;j_{L}=j_{lb}+j_{sh}$$

Therefore, a system BPC under sunlight can be expressed as:(4)$$\begin{array}{c} \;j_{i}=j_{L}+j_{C}\approx j_{b}+j_{sh} \end{array}$$

This is experimentally proved to be true recently in a prototypical BPE material BaTiO${}_{3}$ [40]. As aforementioned that the Voc in these ferroelectric perovskite materials is not limited by the bandgap owing to extracted photocarriers. The main obstacle of achieving high conversion efficiency lies on the photocurrent response. Equation (Equation 4) proves the possibility of using just shift current and ballistic current, which originated from the asymmetric carrier generation, to express the total photocurrent. Therefore as shown in Figure 2, we designed an experiment, based on the different electric hysteresis (P–E) loops of the pure BFO and co-doped BGFMO showing different structural polarization, we can simply verify the BPE by measuring the BPC components with standard J-E curves. Figure 2a shows experiments performed at room temperature under polarization-modulated commercial 405 nm laser illumination using pure and co-doped BFO films. Photovoltaic currents $j_{L}$ and $j_{C}$ were represented through leakage currents collecting from a polarizer before passing through a wave plate, λ/2 or λ/4 (Thorlabs). Figure 2b from left to right are illustrating the measured photo-respond leakage current of BFO, BGFMO-4 and BGFMO-8. The perovskite structure was distorted more with Gd and Mn co-doping showing from the inset of Figure 1a. Whereas in Figure 2c the electric hysteresis (P–E) loops of the pure BFO and co-doped BGFMO thin films were measured at 10 kHz at room temperature. The inset of Figure 2c shows a piezo-response force microscopy, displaying the out-of-plane morphology of the ferroelectric domains under an applied voltage of 10 V. The domain inversion phase difference is distinguished by different colors and is almost uniformly distributed, which likewise, has a similar size to the morphology shown in Figure 1b–f and thus indicates the origins of polarization [41]. The remnant polarization from pure BFO thin film at 0.5 µC/cm${}^{2}$ increases to BGFMO-8 at 13.3 µC/cm${}^{2}$. This increase could be attributed to a larger off-center ion in the Fe-O octahedral due to Gd${}^{3+}$ and Mn${}^{3+}$ ions having smaller ionic radius compared with Bi${}^{3+}$ and Fe${}^{3+}$, and thus yielding a larger ferroelectric polarization in the co-doped perovskite films.

Figure 3a–e shows the J-E curves as λ/4 plate rotates for BGFMO-4, BGFMO-8 and BFO, respectively. We have measured all samples with each multiple times (Appendix A). Among results, there is a surge increase of light induced current in BGFMO-8. And we can observe the concurrently increasing of either linearly or circularly polarized light induced photo current from Figure 3f. This can lead to two inferences: (i) BGFMO-8 sample is indeed the best one among these samples in terms of photocarrier generation. (ii) Shift and ballistic current is indeed in effect as the origin of $j_{L}$ and $j_{C}$ analyzed above. Noted this is an indirect way of measuring the shift current but it makes a simple “proof of principle” type new method that by one-step measuring of the leakage current, one can prove the existence of BPC. The insets from (a–e) of amplified parts clearly show the polarized light arose more carriers, especially in more significant non-centrosymmetrical ones. Moreover, recent studies revealed that the shift and ballistic current can generate longer traveling photo-carriers [42,43], which is distinct from the conventional p-n junction solar cells and also strengthened our proposals. Figure 3f shows the variation of photo current as rotation of λ/4 plate based on BGFMO-8, and Figure 3g shows the curves of photo current from BFO to BGMFO-12 as codoping increasing, indicating the same trend as in Figure 5b with BGFMO-8 having the largest $j_{i}$. While BFO has few photo current, and the sum of ballistic and shift current is the total $j_{i}$, as indicated from Equation (Equation 4).

### 3.3. Magnetic Photogalvanic Effect

We can consider further the effect of the built in magnetic field $j_{H}$. The magnetic hysteresis loops are measured in a Quantum Design Dynacool cryostat at room temperature and magnetic fields between −9 T and 9 T shown in Figure 4a. All the samples from BFO to BGFMO-12 demonstrate a ferromagnetic behavior as also can be seen from the inset amplified figure showing a ferromagnetic field of 11.26, 13.97, 28.68, 62.82 and 46.60 emu/cm${}^{3}$, respectively. Compared with pure BFO, the magnetization of the doped samples was significantly enhanced. Similar to the ferroelectric and leakage current trend as also shown in Appendix A. BGFMO-8 film exhibits the strongest magnetization with a saturation magnetization of 62.28 emu/cm${}^{3}$. This is normally attributed to the increase of the spin canting angle or the suppression of spiral spin owing to structure transformation from the doping [32]. Additionally though, the structural phase transition here discovered in Ga–Mn co-doped BFO thin films yields releasing locked magnetization resulted from collapse of space modulated spin structure as shown in Figure 2c, which thus shares the same origin of ferroelectricity.

To verify the coupling between ferroelectricity and ferromagnetism, we tested the magnetoelectric coupling coefficients α of different samples by a self-made system. α is essentially the stress induced by a magnetostriction caused electric field, which resulted from the magnetic field transmitting to the piezoelectric phase by means of the piezoelectric effect. Not like in most heterojunction composite cases, here in BGFMO the deformation caused by built in magnetic field will yield a large number of positive charges accumulated at the Pt electrode, and then forming a polarization electric field and thus shortens the energy band. Finally, it promotes photo-induced electrons and holes moving towards the positive and negative electrodes as is shown in Figure 4b, the coupling coefficients increase first and then decrease. The maximum coefficient 1.32 mV/cmOe comes from BGFMO-8, which again, shares the same trend as exhibited in Figure 5b. It has been proved that the antiferromagnetic materials can also produce DC current without external magnetic field, which is called magnetic photogalvanic effect [44]. The key is to break the momentum-inversion symmetry in the band structure. And our correlations proved here can provide guidance and simple methods to design and testify more photoconductive magnetoelectric multiferroic materials.

## 4. Discussion

The trend should be similar if we go further to observe the decrease of bandgap along codoping Gd and Mn shown in Figure 5a, which represents a calculated Tauc equation with a direct bandgap. The optical bandgap gradually decreases as the increase of the co-doping shown in Figure 5b. This can be mainly attributed to the rearrangement of molecular orbital and distortions of the angle of Fe-O-Fe bond while doping [45]. Noted for its application in solar cells that the short-circuit current depends on the shift current and ballistic current conductivity in our non-central-symmetrical type of materials. Therefore we can estimate the open-circuit field of BGFMO-8 with a bandgap of 1.97 eV in the case of shift current density dominance as in Figure 3d. With a finite electron-phonon interaction impurity band transition process, $E_{OC}=J_{SC}$/$\sigma^{DC}$, where $\sigma^{DC}$ is the sample’s conventional DC conductivity. Using a shift current density of 1.2 A/m${}^{2}$ for BFO film under sun light and a small bulk conductivity $10^{-2}$(Ωm)${}^{-1}$ [46], we estimate the open-circuit field to be 120 V/m. As for a high quality sample, the open-circuit voltage can indeed be much higher than the bandgap, and it would be possible to take advantage of the non-central symmetrical ferromagnetic materials shift and ballistic current to be applied in solar cells.

AT this stage, one may tend to summarize that the photo-carrier respond is proportional to the degree of asymmetry, especially in these all-inorganic perovskite multiferroic devices, as can be seen in Figure 2b with the three meters displaying increasing photocurrents along with larger polarization. However, it is still non-secure to make such a claim, as how much quantity exactly of asymmetry variation on enhancing BPE is still elucidated. And more importantly, the mechanisms of the effects that give rise to high Voc and their connection with shift and ballistic currents are still in debate [47]. Therefore, here we only demonstrated a simple method to show the mechanism of bulk photovoltaic current in all-inorganic perovskite multiferroic materials. What is more, we also illustrated that for more polarized samples such as in Figure 3c, the circular polarized light had larger effect photo-induced carriers. Tremendous amount of research are necessary for further discoveries in this field.

## 5. Conclusions

Bi${}_{0.85}$Gd${}_{0.15}$Fe${}_{1-x}$Mn${}_{x}$O${}_{3}$ perovskite thin films were prepared on both Pt/Ti/SiO${}_{2}$/Si and ITO glass substrates using a sol-gel process. The effects of co-doping on the electrical and optical properties were discussed. The experimental results showed that a trend, either in multiferroic or optical point of view can be regulated, which brought in a simple method of using leakage I-V cutrves to describe the mechanisms of bulk photovoltaic current. The relationship between each properties was discussed, providing a feasible way for the application of the all-inorganic multiferroic perovskite materials in the photoelectric field.

## Figures and Tables

**Figure 1 nanomaterials-13-00429-f001:**
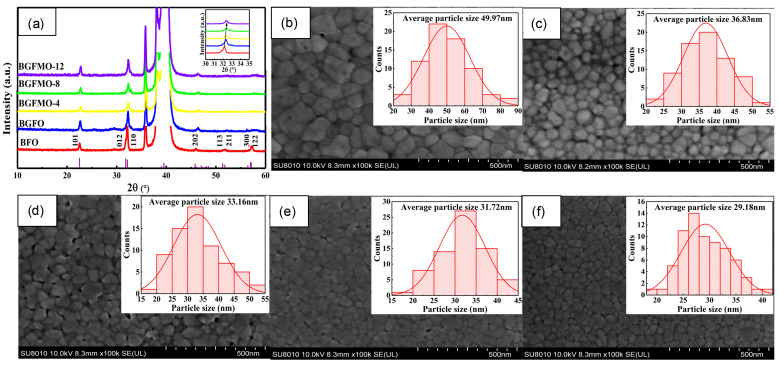
(**a**) XRD spectrum from BFO to BGFMO-12. The inset shows the redshift peak of (012) and (110). (**b**–**f**) SEM micrographs of (**b**) BFO, (**c**) BGFO, (**d**) BGFMO-4, (**e**) BGFMO-8 and (**f**) BGFMO-12 films. The insets: Histograms of particle size distribution of different films.

**Figure 2 nanomaterials-13-00429-f002:**
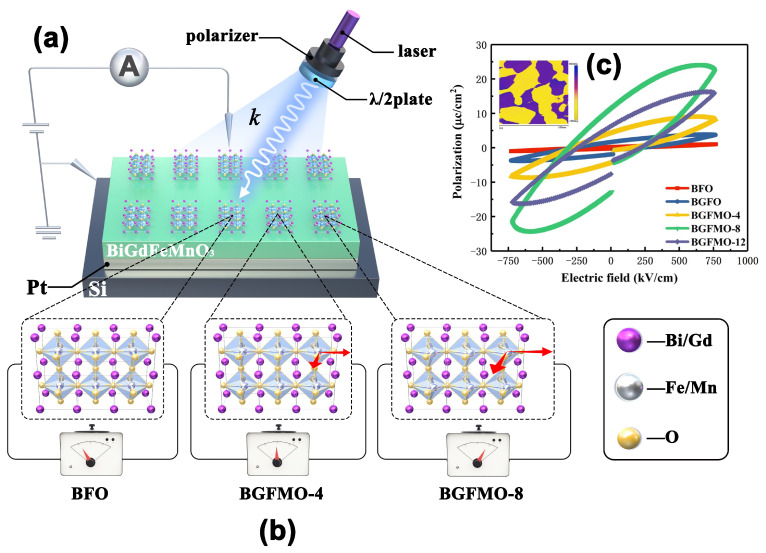
(**a**) Designed measurements of photoresponse under laser illumination, on samples of BGFMO on Pt/Si substrate. Ferroelectric test system is connected on the top and bottom electrode, with the blue arrow denote light propagation of wave vector k. (**b**) Illustration of increasing distortion/polarization (indicated by red arrows) affects on BFO, BGFMO-4 and BGFMO-8, respectively. (**c**) Electric hysteresis (P–E) loops of the pure BFO and co-doped BGFMO thin films. The inset shows a PFM measurement with a 0.5 μm × 0.5 μm of BGFMO-8s.

**Figure 3 nanomaterials-13-00429-f003:**
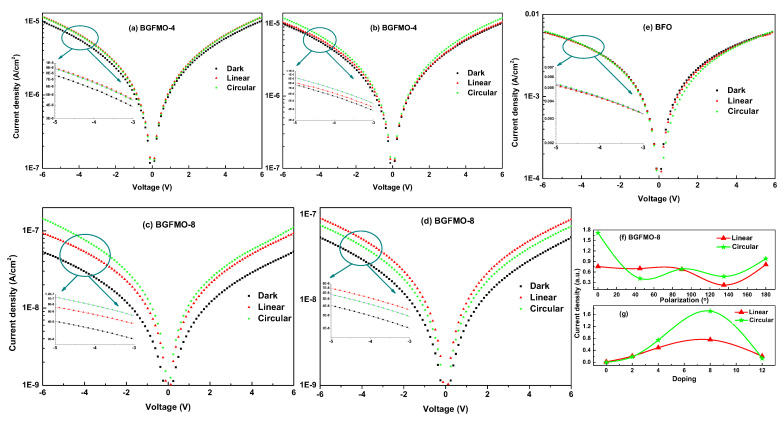
(**a**–**e**) Leakage I-V curves measured at room temperature under polarization-modulated commercial 405 nm laser illumination using pure BFO and co-doped films. (**f**) Variation of leakage current as rotation of λ/4 plate based on BGFMO-8. (**g**) The trend of linearly and circularly polarized photo current from BFO, BGFO to BGFMO-12.

**Figure 4 nanomaterials-13-00429-f004:**
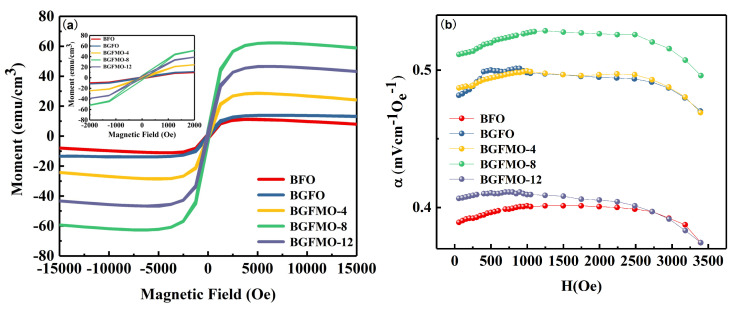
(**a**) Band gap of different film samples. (**b**) The relationship curves between Pr, Ms and E${}_{g}$.

**Figure 5 nanomaterials-13-00429-f005:**
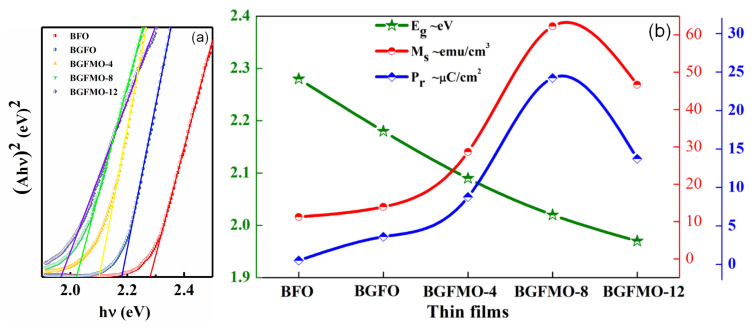
(**a**) The magnetic hysteresis loops of BFO, BGFO, BGFMO-4, BGFMO-8, and BGFMO-12 thin films. (**b**) The magnetoelectric coupling coefficients of these samples.

## Data Availability

The data that support the findings of this study are available from the corresponding author upon reasonable request.

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
