# Peer review of "Bulk Photovoltaic Current Mechanisms in All-Inorganic Perovskite Multiferroic Materials"

_nanomaterials, 2023, doi:10.3390/nano13030429_

Round 1

Reviewer 1 Report

The topic is interesting to the readers,yet the paper needs to be revised, I recommend Major revision. 

The author must portray clearly about "One type Proof method"..

How the author's are measuring leakage current?

Methodology to detect leakage current and it's science behind it must be clearly addressed.

The novelty of the proposed work is still not clearly visible at the end of introduction 

Reviewer 2 Report

The authors have investigated the electrical and optical properties of  perovskite thin films prepared on ITO glass substrates using a sol-gel process.

The manuscript appears  well- structured. The english needs to be improved in order to easy interpret and understand the article ; for example in row 41  and in row 51

Perovskite solar cells (PSCs) are a class of substances with the typical chemical formul

This sentence needs to be written again  !!!

All-inorganic halide perovskites are considered as  of significant potential since 

This sentence needs to be written again  !!!

The Figure 1 needs to be magnify because the axis and resolution appear unclear!

I suggest to extend the introduction considering to add/cite the other kind of  solar cells methodology well-studied in 

SCIUTO, Grazia Lo, et al. Organic solar cells defects detection by means of an elliptical basis neural network and a new feature extraction technique. Optik, 2019, 194: 163038.

Lo Sciuto, Grazia, et al. "Organic solar cells defects classification by using a new feature extraction algorithm and an EBNN with an innovative pruning algorithm." International Journal of Intelligent Systems 36.6 (2021): 2443-2464.

Reviewer 3 Report

The authors reported the effects of Gd and Mn co-doped in the inorganic perovskite thin films prepared by a sol-gel process on their microtopography, grain boundries, multiferroic, and optical properties were studied. Based on the author's founding, the reviewer suggests the publication of this manuscript to Nanomaterials after resolving the following issues.

1) The authors should provide the stability of the cells.

2) As the ferroelectric materials were introduced in the cell, the electrical resistance can be increased. What is the advantage of the suggested system) The authors enhances the novelty of this system.

Round 2

Reviewer 1 Report

The revision Made by the author's is appreciated